# ESM-Effect: An Effective and Efficient Fine-Tuning Framework towards Accurate Prediction of Mutation's Functional Effect

## Abstract

Functional effect prediction of mutations, especially for properties like catalytic activity, holds greater significance for clinicians and protein engineers than traditional pathogenicity predictions. Recent approaches leveraging static ESM1 embeddings or multimodal features (e.g. embeddings, structures, and evolutionary data) either (1) fall short in accuracy or (2) involve complex preprocessing pipelines. Moreover, functional effect prediction suffers from (3) a lack of standardized datasets and metrics for robust benchmarking. We address these challenges by systematically optimizing ESM2-based functional effect prediction: Through extensive ablation studies, we demonstrate that fine-tuning significantly outperforms static embeddings, scaling laws for model size are non-transferable and LoRA matches full fine-tuning performance, deviating from trends observed in natural language processing. Our framework, ESM-Effect, fine-tunes 35M ESM2 layers with an inductive bias regression head achieving state-of-the-art performance. It slightly surpasses multimodal competitor PreMode indicating redundancy in structural and evolutionary features. We further propose a benchmarking framework featuring robst test datasets and strategies, and the relative Bin-Mean Error (rBME), as a metric designed to emphasize prediction accuracy in challenging, non-clustered, and rare gain-of-function regions. rBME better reflects model performance compared to commonly used Spearman's rho, as evidenced by improved plot-based analyses. As ESM-Effect exhibits mixed transferability to different unseen mutational regions, we identify multiple areas for improvement such as finer-grained pretraining strategies.

## 1 Introduction

Accurate prediction of mutation effects remains a central challenge in computational biology, as mutations exhibit heterogeneous impacts on health and disease. This challenge is further exacerbated by the rapid increase in mutations identified in routine patient sequencing, driven by the decreasing cost of sequencing technologies (Pasmans et al., 2021). While Deep-Mutational Scans (DMS, i.e. measuring a specific property of all possible mutations in a given protein) offer clinicians precise functional insights, they are laborious, expensive and rare, often failing to cover the full protein of interest (Karczewski et al., 2020). These limitations underscore the need for accurate computational methods to efficiently predict the functional effect of mutations.

With the advent of artificial intelligence, advanced deep learning models (Krizhevsky et al., 2017) join the traditionally machine-learning-dominated landscape of mutation prediction (Ioannidis et al., 2016; Adzhubei et al., 2010). The current landscape is characterized by two axes (cf. Figure 1):

- **(a)** whether the mutation effect is predicted as a unidirectional pathogenicity score or a bidirectional functional effect (i.e., increasing or decreasing a specific property or activity) and
- **(b)** whether the model performs classification or regression.

Most existing models focus on pathogenicity prediction (i.e. how physiological or wildtype-similar a mutation is) and use regression-based approaches. These models adopt a generalist strategy, scor-

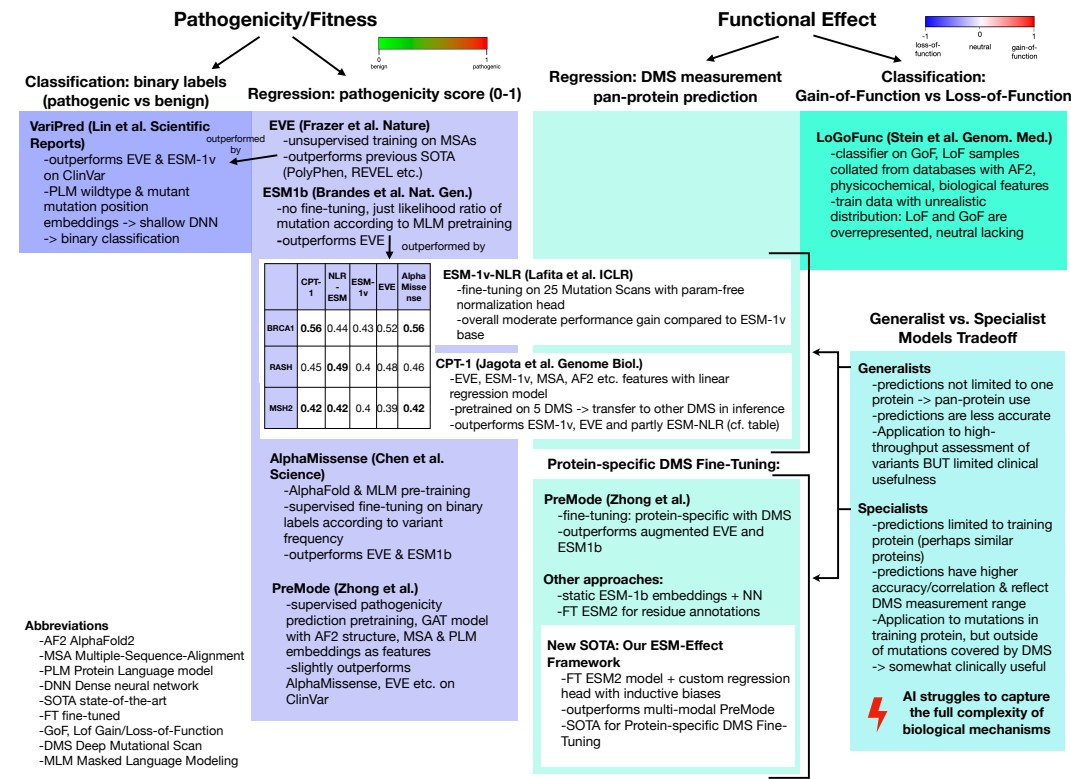

Figure 1: Survey of existing methods illustrating the trade-off between broadly applicable but less precise models and highly precise models limited to their training protein. Notably, the latter can produce high-quality predictions only for mutations within the same protein as the training DMS. Despite this limitation, such models remain valuable, as DMS datasets typically focus on specific protein domains and often contain incomplete data due to failed mutagenesis experiments.

ing all possible variants across the (human) proteome . This enables high-throughput screening and facilitates proteome-wide mapping (Cheng et al., 2023). However, pathogenicity predictors — whether trained on multiple DMS datasets, ClinVar annotations or physiological sequences — struggle to accurately predict the bidirectional functional effects of specific mutations, such as rare gain-of-function enzyme mutations. This limitation arises from the biological complexity and specificity required for such tasks, which cannot be reliably captured by large-scale pretraining and the current architectures (Livesey & Marsh, 2023). However, clinical decision-making often depends on understanding the precise functional effect of mutations (i.e. increase/decrease of a specific protein property) (Iyer et al., 2023).

In this paper, we address these limitations by

- (1) first evaluating the shortcomings and potential of existing methods for both pathogencity and functional effect prediction and

- (2) then developing the optimal framework for ESM2-based functional effect prediction through detailed ablations of various fine-tuning strategies and prediction head architectures. Based on these insights, we propose the **ESM-Effect framework**, which achieves state-of-the-art (SOTA) performance on functional effect predictions outperforming multimodal competitors.

- (3) Finally, we analyze the strengths and weaknesses of ESM-Effect's capabilities and propose robust benchmarks to facilitate further progress in the field.

## 2 BACKGROUND

**Mutation Effect Prediction as a question of pathogenicity** Mutations affect proteins in diverse ways, making precise measurement of their impact challenging. To simplify, the concept of "mutation pathogenicity" categorizes mutations as either "pathogenic" (disrupting physiological protein function) or "benign." Pathogenic mutations typically reduce organism fitness and are rare in natural sequences, such as those in UniRef (Suzek et al., 2007), representing the physiological sequence space. Models can learn pathogenicity from large datasets of natural sequences, scoring the likelihood of mutations based on their presence in (physiological) evolutionary or MSA data (Meier et al., 2021). However, this broad definition oversimplifies the diverse effects mutations can have. For example, pathogenic mutations in an ion channel might either increase or decrease affinity (Kullmann & Hanna, 2002), whereas pathogenic mutations in collagen disrupt its fibrillary structure (Dalgleish, 1997).

**Mutation Effect Prediction as a question of functional effects** In contrast, functional effect prediction considers a wider range of impacts, such as catalytic activity, binding and stability, which are more directly applicable to precision medicine and protein engineering. However, achieving high accuracy requires both protein-specific supervised data (Zhong et al., 2024) and appropriate architectures (incl. training strategies).

## 3 RELATED WORK

### 3.1 PROTEIN MODELING AND PATHOGENICITY PREDICTION

Methods like AlphaFold2 (AF2) predict protein structures from MSAs, capturing evolutionary information about residue interactions (Jumper et al., 2021) and Transformer-based Protein Language Models (PLMs), like ESM-1b and ESM2, learn protein semantics by predicting masked amino acids from evolutionary sequences (Rives et al., 2021; Lin et al., 2023; Rao et al., 2020). As these models learn sequence and structure physiology they be directly applied to predict the lack thereof in form of the likelihood ratio of a mutant and wildtype residue (e.g., AlphaMissense, EVE building on MSAs (Cheng et al., 2023; Frazer et al., 2021) and pretrained PLMs like ESM-1v (Meier et al., 2021; Brandes et al., 2023)). Some methods refine predictions using DMSs, which offer sufficient signal for pathogenicity despite heterogeneous properties across different DMSs. Examples include fine-tuning ESM-1v on 25 DMSs with a Normalized Log-odds Ratio (NLR) head (Lafita et al., 2024) and combining EVE, ESM-1v, and AF2 features in a regression model (Jagota et al., 2023). However, these methods struggle with multi-directional functional effects, particularly for Gain-of-Function mutations in DMSs like SNCA (Livesey & Marsh, 2023). In summary, while pathogenicity models effectively distinguish benign and pathogenic mutations, they fall short in predicting multi-dimensional functional effects as demonstrated in the **Appendix** 7.1.

### 3.2 MODELS FOR FUNCTIONAL EFFECT PREDICTION

To address functional effect prediction, existing models extend pathogenicity predictors: Derbel et al. (2023) and Marquet et al. (2022) use static ESM embeddings combined with a neural network head to predict functional effects from DMSs. Saadat & Fellay (2024) fine-tune ESM2 for residue-level protein sequence annotation (e.g., identifying functional features like active sites) and then classify mutations based on the probability difference of annotated features between reference and mutant sequences, comparing this to ClinVar labels rather than DMSs. LoGoFunc, another method, performs three-class classification using a diverse feature set to make genome-wide predictions (Stein et al., 2023).

Studying the extent of the expected benefit of fine-tuning PLMs, Schmirler et al. (2024) showed that ESM2 fine-tuned with Low-Rank-Adaptation and a neural network regressor on top of the mean mutant embeddings outperforms the simple, Non-PLM baselines Homology-Based Inference and the statistical model Reference Free Analysis (RFA) on three DMS (AAV, GFP and GB1).

The latest and most complex model for functional effect prediction is PreMode (Zhong et al., 2024; Zhong & Shen, 2022), which is pretrained on 4.7M pathogenicity-labeled mutations and then fine-tuned on a specific DMS. PreMode uses the static wildtype embeddings (650M ESM2 model),

MSAs and additional mutation-specific features as node vectors and the AF2-predicted structure as a distance matrix for a star graph attention model. PreMode outperforms a Random Forest model, pretrained 650M ESM2 embeddings with a single layer perceptron and other state-of-the-art methods given the same input features as PreMode (e.g. EVE). Besides, the authors' preliminary analyses showed that LoF, GoF and neutral mutations have distinct but overlapping (i.e. no unique intervals exclusive to any one class) distributions for pLDDT scores, conservation levels, and solvent accessibility.

Finally, pathogenicity predictors like CPT-1 and ESM-1v NLR can also be used for functional effect prediction, but their accuracy is limited due to their generalist nature.

## 3.3 Databases and existing Benchmarks for Mutation Effect Prediction

To advance and compare pathogenicity predictors, large databases of annotated mutations, Deep Mutational Scans (DMS) and clinical annotations have been developed as well as numerous experimental efforts exploring and testing mutations in the wet lab (Backman et al., 2021; Dunham & Beltrao, 2021; Esposito et al., 2019; Exome Aggregation Consortium et al., 2016; Gao et al., 2023; The UniProt Consortium et al., 2023). Notable resources include ProteinGym, which serves both as a repository for Deep Mutational Scans (DMS) and as a benchmarking platform for evaluating the latest pathogenicity predictors (Notin et al., 2023). Similarly, MaveDB provides a curated repository of DMSs, while ClinVar includes clinical annotations with benign and pathogenic labels (Landrum et al., 2018; Rubin et al., 2021).

Livesey & Marsh (2023) used 26 DMS to benchmark 55 pathogenicity predictors reporting respectable performance (measured by Spearman correlation and AUROC) in distinguishing pathogenic variants. However, their findings underscore substantial variability across predictors, with particularly poor performance on DMSs that included gain-of-function (GoF) mutations.

## 4 ESM-Effect

### 4.1 Problem Statement

As **existing methods** either do **not fine-tune ESM2**, only **use static embeddings** or different regression heads, we begin the development of ESM-Effect by detailed ablations of combinations of different training regimen and regression heads. Thereby, we hope to distill the most performant characteristics of existing approaches into ESM-Effect which we then compare to the multi-modal PreMode model which uses embeddings, AF2 structure and MSAs to assess the benefit of multi-modality.

### 4.2 ESM-Effect: Developing the optimal prediction architecture

**ESM2 Model Size** Scaling laws in natural language processing (NLP) suggest that larger models are more compute-efficient for modest datasets (Kaplan et al., 2020). These principles have also been shown to hold in biological applications, with increasing ESM2 model size leading to lower language modeling loss and better performance structure prediction (Lin et al., 2023). To investigate whether these trends extend to the downstream task of functional effect prediction, we evaluated ESM2 models of varying sizes on AAV, GB1, and GFP DMS datasets (models trained by Schmirler et al. (2024)) along with the validation perplexity reported by Lin et al. (2023) (cf. Figure 2), finding that scaling laws do not hold in this context. No obvious performance improvements emerge with larger models across all DMS uniformly, and we observe comparable results across model sizes. Consequently, we select the 35M ESM2 model due to its favorable balance of computational efficiency and performance.

**The Value of Fine-Tuned Embeddings** Previous approaches to functional effect prediction have relied on static embeddings from fully frozen ESM models combined with various prediction heads (Marquet et al., 2022; Derbel et al., 2023; Zhong et al., 2024). To evaluate whether this limitation constrains performance, we compare static 35M ESM2 embeddings to fine-tuned 35M ESM2 embeddings (with the last two layers unfrozen) across four DMS datasets. Both approaches use a prediction head that inputs the mean of the mutant embeddings into a Single-Layer Perceptron

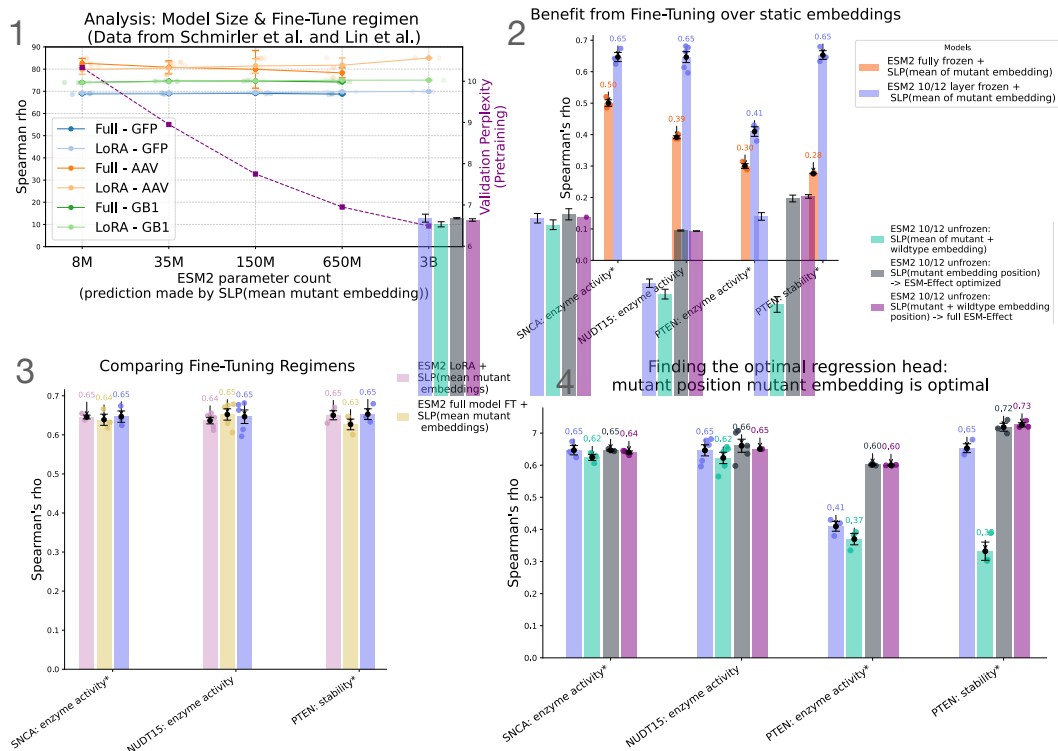

Figure 2: 1 - Kaplan et al. (2020) scaling laws do not extend to downstream functional effect fine-tuning performance but are consistent with pretraining metrics (e.g. validation perplexity, CASP14 performance). Minimal performance difference between fine-tuning regimens (LoRA vs. full Fine-Tuning). 2 - Significant benefit from fine-tuned embeddings. * indicates that only three of five available seeds were used due to resource limitations. 3 - Minimal performance differences between Fine-Tuning strategies. Unfreezing the last two layers was selected for ESM-Effect due to interpretability advantages etc. Information on training characteristics for the PTEN DMS is in the Appendix 7.7. 4 - Analysis of the optimal regression head. Note that mutation position based heads require maximal 10 epochs for optimal performance while mean based heads take longer and suffer from instable training for PTEN DMS (cf. Appendix 7.7)

(SLP) for a fair comparison. As shown in Figure 2, fine-tuned embeddings consistently outperform static embeddings, despite dataset-specific variations. These results point out a critical shortcoming of existing methods and establish fine-tuning as a key design choice for ESM-Effect.

**LoRA vs Full vs Partly Fine-Tuning** Our previous analysis of the data from Schmirler et al. (2024) also demonstrated that LoRA and full fine-tuning achieve comparable performance. To independently validate this and extend the analysis, we evaluated LoRA, full fine-tuning and partial fine-tuning (unfreezing the last one or two layers) on three diverse DMS datasets. As shown in Figure 2, all three strategies performed equivalently. This result diverges from findings in NLP tasks, where LoRA has been shown to underperform full fine-tuning in domains like programming and mathematics (Biderman et al., 2024). Accordingly, the functional effect prediction task exhibits unique characteristics, making LoRA and layer-freezing viable alternatives for parameter-efficient fine-tuning within the ESM-Effect framework. For further development, we selected the strategy of unfreezing the last two layers for ESM-Effect due to its reduced need for extensive hyperparameter tuning and improved interpretability (cf. Appendix refsec:ablation).

**Regression head** With the optimal model size and fine-tuning strategy determined, we subsequently evaluated the optimal regression head for the ESM-Effect framework. Previous methods have primarily used either the mean embedding of the mutant sequence or combined static embeddings of the mutant and wildtype sequences at the mutation position as input to a feed-forward neural network. Building on fine-tuning the 35M ESM2 model (with 10 of 12 layers frozen), we evaluated

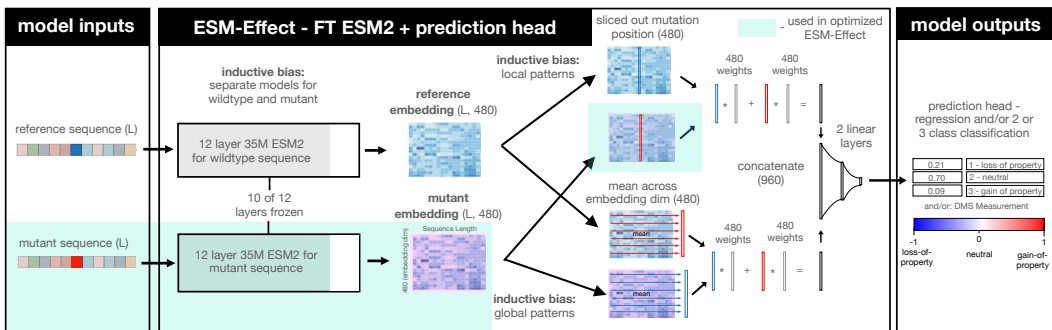

Figure 3: Architecture of full ESM-Effect. Embedding parts and data used for optimized ESM-Effect is highlighted in light green.

four regression head designs across four DMS datasets: (1) The mean embedding of the mutant sequence, (2) a linear combination of the mean embeddings of mutant and wildtype sequences, (3) the embedding at the mutation position of the mutant sequence and (4) a linear combination of the mutation position embeddings of mutant and wildtype sequences.

This analysis allowed us to assess (1) the relative importance of the mutation position and (2) the specific wildtype residue as references to the physiological sequence space. As shown in Figure 2, while all four regression heads performed similarly for SNCA and NUDT15 DMS datasets, the mutation position-based regression head significantly outperformed mean-embedding-based approaches for the PTEN stability and PTEN enzyme activity DMS datasets. Notably, this performance gain occurred even though the second mean-based approach incorporated information about the mutation position and wildtype residue, showing the utility of the mutation position as a valuable inductive bias for these tasks.

**The ESM-Effect architecture** comprises the 35M ESM2 model with 10 of 12 layers frozen and a neural network regression head. This regression head processes the mutant and wildtype sequence embedding at the mutation position (cf. Figure 3).The model's performance is driven by two key **inductive biases** in the regression head:

- the mutation effect is relative to a wildtype sequence
- mutation impact is largest at the mutation position

While the full architecture, incorporating both mutant and wildtype embeddings, directly implements these biases, a simpler variant — using only the mutation position embedding of the mutant sequence — achieves comparable performance with approximately half the computational cost. We term this streamlined version the **optimized ESM-Effect** model, as it encapsulates both inductive biases in a minimal and efficient form.

## 5 RESULTS

### 5.1 PERFORMANCE COMPARISON: OPTIMIZED ESM-EFFECT OUTPERFORMS EXISTING SOTA METHOD PREMODE

Next, we compare ESM-Effect to the state-of-the-art method, PreMode, which is pretrained on millions of pathogenic variants and fine-tuned on nine diverse DMSs. Unlike ESM-Effect, which relies solely on sequence input and its learned embeddings, PreMode incorporates static ESM2 embeddings, AF2 structures, and multiple sequence alignments (MSAs). Given the significant performance gains that multimodal approaches achieve in the natural language domain, we anticipated PreMode to outperform ESM-Effect. However, PreMode's ablation analysis reveals only a marginal performance drop when any one of the three modalities is excluded, indicating that the information they provide for functional effect prediction is largely redundant.

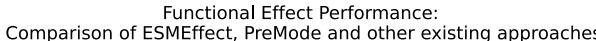

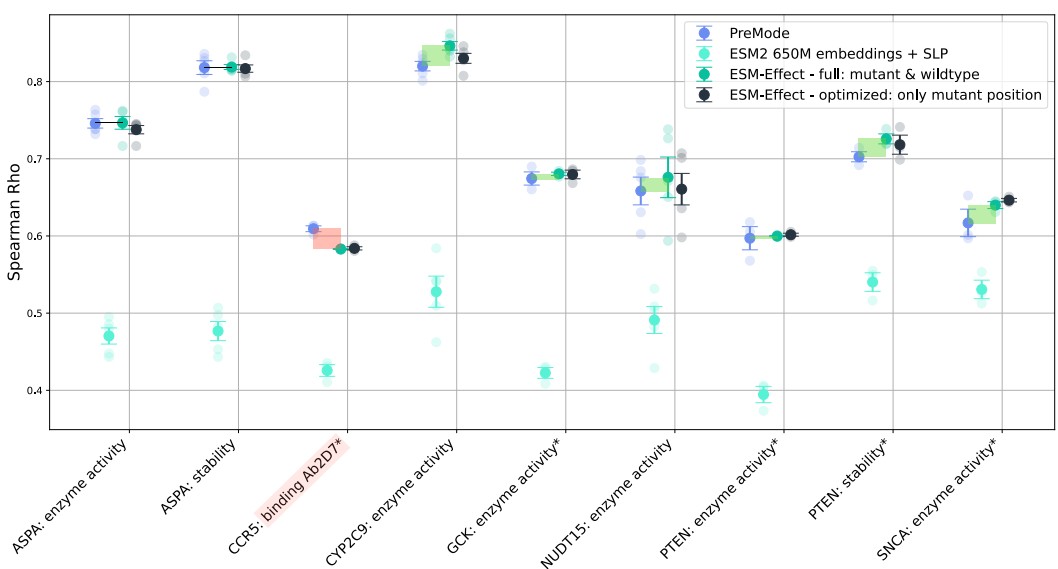

Figure 4: Performance Comparison of ESM-Effect with multi-modal PreMode. Stars indicate ESM-Effect mean performance on the same five 80-20 train-test split seeds as PreMode.

**Indeed, optimized 35M ESM-Effect performs slightly better than PreMode despite having two input modalities less** (cf. Figure 4, Table 1). ESM-Effect models almost always outperform PreMode by varying margins except for the DMSs measuring mutation impact on CCR5 antibody binding which suggests that PreMode's knowledge of AF2 structure gives it a competitive advantage because protein structure is involved. The full ESM-Effect model and the optimized model almost always perform on par. This relates to our discussion of the arguable existence of one fixed wild-type sequence in the Appendix and underpins that ESM2's own understanding of the physiological sequence space suffices and it does not require the (or "a specific") wildtype residue as orientation towards to phyiological sequence space. Besides, we also experimented with Test-Time-Training finding mixed improvements (cf. Appendix 7.3) (Bushuiev et al., 2024).

| model

task name | ESM Effect full | ESM Effect optim. | ESM2 10/12 frozen mean | SLP (embed.) | ESM2 LoRA mean | PreMode |
|---|---|---|---|---|---|---|
| ASPA: enzyme activity | **0.747** | 0.738 | | 0.470 | | 0.746 |
| ASPA: stability | **0.819** | 0.817 | | 0.477 | | 0.818 |
| CCR5: binding Ab2D7* | 0.583 | 0.584 | | 0.426 | | **0.609** |
| CYP2C9: enzyme activity | **0.846** | 0.830 | | 0.528 | | 0.820 |
| GCK: enzyme activity* | **0.680** | **0.680** | | 0.422 | | 0.674 |
| NUDT15: enzyme activity | **0.676** | 0.661 | 0.646 | 0.491 | 0.636 | 0.658 |
| PTEN: enzyme activity* | 0.600 | **0.602** | 0.544 | 0.395 | 0.475 | 0.597 |
| PTEN: stability* | **0.726** | 0.718 | 0.653 | 0.540 | 0.650 | 0.703 |
| SNCA: enzyme activity* | 0.640 | 0.646 | **0.647** | 0.531 | 0.646 | 0.617 |

Table 1: Table comparing the mean spearman rho on DMS between ESM-Effect models, PreMode and other setups on 3 or 5 seeds. Mean models use the mutant sequence only.

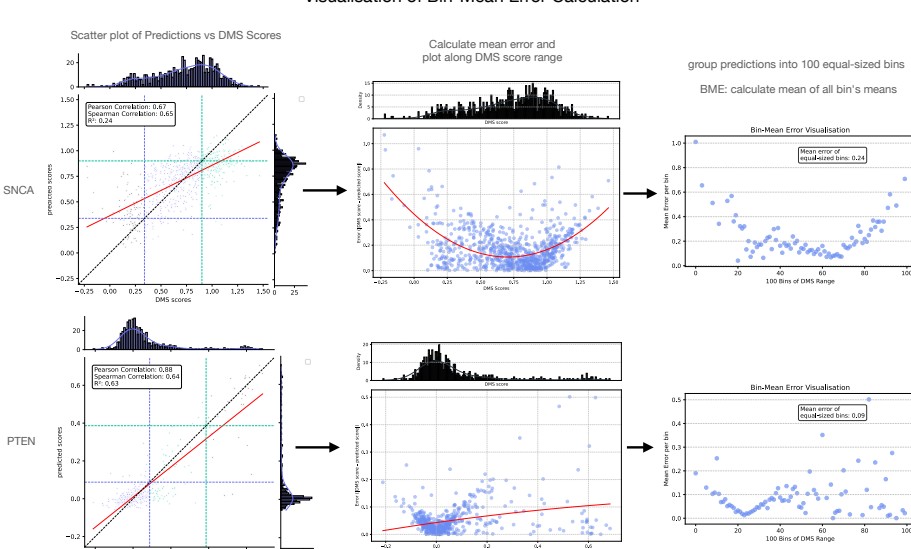

Figure 5: Visualization of the BME calculation steps. Predictions stem from LoRA ESM2 + SLP(mutant embeddings) fine-tuned on SNCA seed 0 and PTEN seed 1 for 20 epochs.

## 5.2 BENCHMARKING FRAMEWORK FOR FUNCTIONAL EFFECT PREDICTION

**General Remarks** While established benchmarks, such as the ProteinGym, exist for pathogenicity prediction, uniform benchmarks including reliable metrics and standardized testing datasets for functional effect prediction are lacking hampering useful comparisons and impeding progress in the field. To address this bottleneck, we propose datasets, including train-test splits, evaluation metrics, and visualizations, to provide a more realistic framework for assessing functional effect predictors. Thus, we encourage future research to adopt and build upon this framework.

**Datasets** We trained and benchmarked ESM-Effect on the same 9 DMS datasets and corresponding test splits used by PreMode, ensuring 1:1 comparability. In previous work, score calculation methods — such as normalization and aggregation of DMS experiment replicas — have often been unclear, as have decisions regarding the inclusion of wildtype scores and the reference sequence isoform used. Standardizing on PreMode datasets or ensuring exact sharing of datasets in the field will address these ambiguities.

We further recommend a more rigorous testing regimen: instead of relying on random data splits, we propose evaluating models on DMS mutations from sequence intervals distinct from those in the training data. This approach provides a **more realistic measure of the model's ability to generalize to new biological contexts** (see Section 5.4). For consistency, it is essential to not only share train-test splits but also the full DMS dataset and to standardize testing intervals across studies.

**Metrics: The relative Binned-Mean Error (rBME)** For pathogenicity prediction, general correlation with DMS scores is often evaluated using scale-invariant metrics like Spearman rank correlation, as implemented in the ProteinGym benchmark. Spearman correlation is well-suited for pathogenicity because it evaluates monotonic relationships and is robust to scale differences across DMS score distributions. However, functional effect prediction requires more nuanced evaluation, particularly for rare, biologically significant mutations, which can be overshadowed by the majority of mutations with neutral effects. Standard metrics like Spearman can mask biases, as models often focus on more frequent, neutral mutations.

To address this, we propose the relative Binned-Mean Error (rBME), a metric that evaluates model performance across distinct mutation effect bins, emphasizing accuracy for rare but impactful mu-

tations (cf. Figure 5): Let the DMS scores and predicted scores (of the test set) be denoted as $y_i$ and $\hat{y}_i$, respectively, for $i = \{1, 2, \ldots, N\}$, where $N$ is the total number of test mutations.

Define the relative error for each mutation $i$ as:

$$\text{relative error}_i = \frac{|y_i - \hat{y}_i|}{\max(|y_i|, \epsilon)},$$

where $\epsilon$ is a small constant to avoid division by zero. Next, group the data points into $n_{\text{bins}}$ equal-width bins based on the value range of $y_i$, where $b_k$ represents the $k$-th bin (typically, $n_{\text{bins}} = 100$). While the model effectively learns the true distribution of DMS scores — capturing clustered regions with many neutral mutations and producing realistic predictions — this step is crucial to mitigate metric bias and ensure balanced treatment across all regions, including easy-to-predict clusters and hard-to-predict, wider regions with rare but biologically significant Gain-of-Function mutations. The relative Bin Mean Error (rBME) is given by the mean of the mean error per bin $b_k$ where $|b_k|$ is the number of data points in bin $b_k$:

$$\text{relative Bin Mean Error (rBME)} = \frac{1}{n_{\text{bins}}} \sum_{k=1}^{n_{\text{bins}}} \frac{1}{|b_k|} \sum_{i \in b_k} \text{error}_i,$$

Normalization of absolute error facilitates comparisons across different DMS, whereas the unnormalized BME metric is suitable for cross-model comparisons on the same DMS. While the optimized ESM-Effect achieves comparable Spearman correlations for PTEN and SNCA (0.59 and 0.63, respectively; cf. Figure 6), the scatter plots reveal a stark difference in performance. This discrepancy is accurately captured by the rBME metric, which reflects the disparity (0.87 vs. 1.40).

## 5.3 PREDICTION ANALYSIS

While most previous studies compare prediction performance with a single metric, only plotting predictions vs. ground truth truly reflects performance. Importantly, a realistic plot should have the same scale for DMS scores and predicted scores axes (i.e. be quadratic) and indicate ideal predictions with an angle bisector. Figure 6 compares the prediction characteristics of the optimized ESM-Effect model and the LoRA ESM2 model with a regression head on top of the mean mutant sequence embeddings. The prediction patterns of optimized ESM-Effect and LoRA ESM2 mean have distinct prediction characteristics, especially for PTEN enzyme activity, where it performs worse (cf. Section 5.1).

The prediction patterns on the SNCA DMS correlate with the high metrics (e.g. spearman rho, low BME and rBME): the models can reliably distinguish activity scores in the upper realm of the DMS score distribution from scores in the lower core region (score -0.2 to 0.2). To further investigate the fine-tuning behavior of ESM2 we analyzed the finer-grained number of unfrozen layers (compared to full, 10/12 frozen

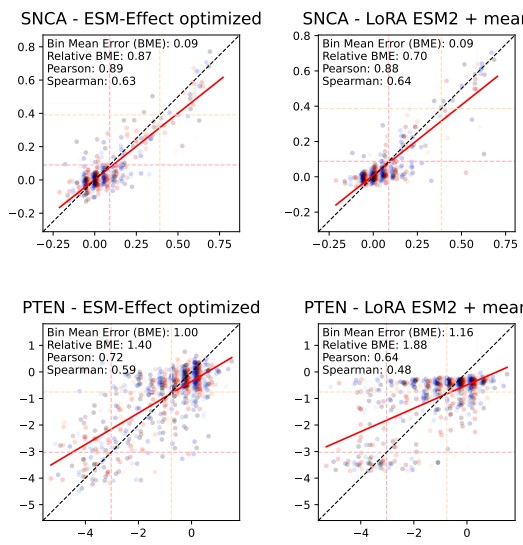

Figure 6: Analysis of optimized ESM-Effect and LoRA fine-tuned ESM2 with SLP(mutant mean embdding).

layers and no fine-tuning above) and the position of one unfrozen layer in the model but none influenced model performance (cf. Appendix 7.2).

## 5.4 Investigating Transfer Capabilities

As part of our proposed benchmarking framework, testing optimized ESM-Effect not by using a random split of the DMS but by using distinct sequence intervals for selecting train and testing mutations assesses generalization: the model has to infer the effect of mutations in the testing interval based on its understanding of the pretraining interval and learned effects from the rest of the protein. We selected SNCA because it features a unique sequence position-to-score relationship as shown in Figure 7. Notably, the last 40 residues are predicted by MobiDB-lite to form a disordered region, lacking stable secondary structure (Necci et al., 2017).

The transfer performance of ESM-Effect is highly dependent on the interval: while the model performs better on intervals enriched with rare, high-score mutations compared to random splits (spearman rho 0.72 vs. 0.65), it struggles within the disordered interval without these mutations (Spearman rho: -0.02). These results show the limitations of current state-of-the-art functional effect prediction models and underscore the challenges in modeling protein regions with distinct structural and mutational properties.

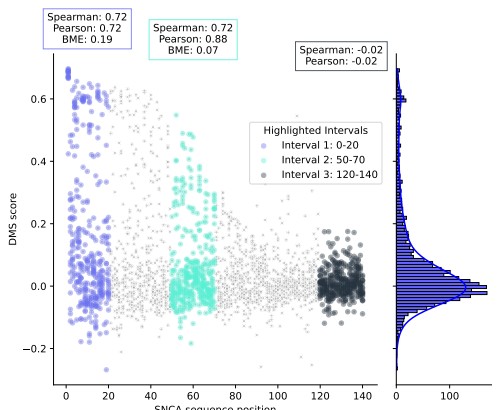

Figure 7: Investigating optimized ESM-Effect's Transfer capabilities on SNCA DMS. Model trained on three random seeds achieves a spearman rho of 0.646. Each testing interval accounts for 14-15% of the total dataset, while the random split used 20%.

## 6 Conclusion

With our step-by-step model development approach building on and improving on previous methods, we develop a new state-of-the-art functional effect predictor: ESM-Effect - an ESM2-finetuning architecture with inductive bias regression head - outperforms SOTA competitors across a range of DMS while sparing structure and MSAs features and focusing on task-specific adaptation of PLM embeddings.

The survey of the pathogenicity and functional effect predictor landscape alongside our analyses reveals shortcomings of current models for a meaningful biological and medical application. The transfer capabilities vary greatly and show that the field of mutation effect prediction has still a long way to go until it can guide treatments and is truly beneficial for real-world applications. We hope to shorten this way with the proposed Benchmarking Framework which emphasizes realistic benchmarking instead of inflated performances and facilitates comparison with future models.

For the downstream task of Deep Mutational Scan (DMS) fine-tuning, our analyses revealed unexpected patterns that diverge from typical natural (and protein) language model scaling behaviors. Notably, test performance remained almost constant across increasing model sizes, and Low-Rank Adaptation (LoRA) consistently matched the performance of full fine-tuning. These observations suggest that the model's utility for DMS prediction may be fundamentally constrained by the limitations of current pretraining approaches. We hypothesize that only low-level, universal knowledge — largely invariant to model size — contributes meaningfully to DMS prediction. The performance plateau indicates that the current pretraining paradigm struggles to capture the nuanced and detailed biological knowledge required for comprehensive mutational effect prediction.

While current pretraining methods are effective in decoding sequence and structural aspects, they seem to fall short in capturing the complex biochemical reactions and interactions of proteins that are only weakly and implicitly encoded in sequence and structure. This suggests the need for new pretraining data sources and objectives (Li et al., 2024), capable of uncovering deeper biological insights to advance the field.

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

# 7 APPENDIX

## 7.1 PATHOGENICITY PREDICTORS PERFORM POOR FOR FUNCTIONAL EFFECTS: ALPHAMISSENSE VS. DMS

Pathogenicity predictors like AlphaMissense carve out the edges of the physiological sequence space, but fall short for accurate functional effect prediction for knowledge of the respective protein's biological mechanism is required (cf. Figure 8)

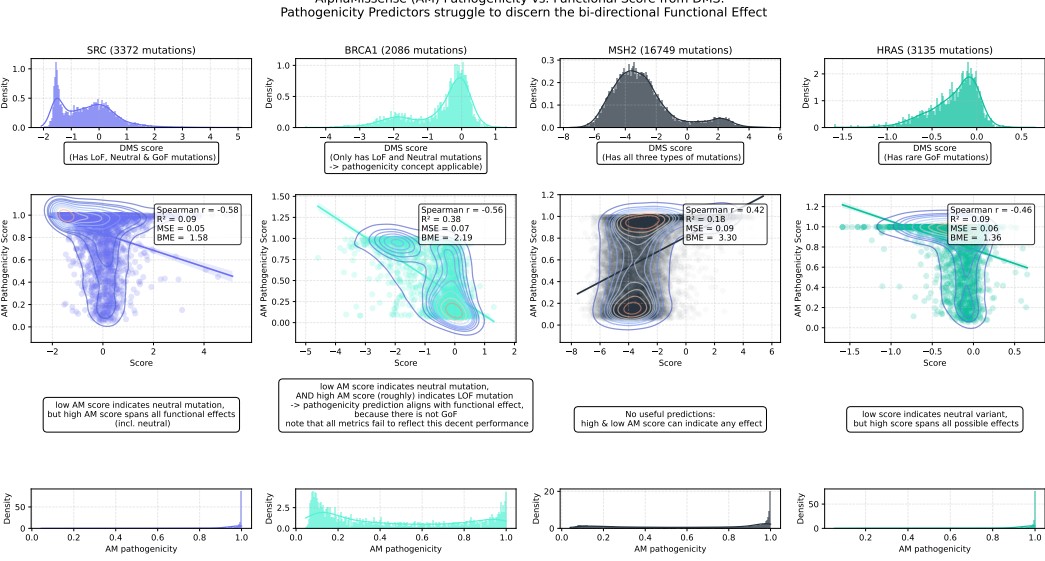

Figure 8: SOTA pathogenicity predictor AlphaMissense on DMS data. Note that the DMSs sometimes not cover the entire protein sequence.

## 7.2 ABLATION AND MODEL ANALYSIS

**Layer Probing** To investigate how the number of trainable layers affects performance, we retrained optimized ESM-Effect with a descending number of layers frozen: the results show that the number of frozen layers has no impact on test performance, as long as at least one layer remains unfrozen, allowing the model to adapt to the specific task (cf. Figure 9). Given that a single unfrozen layer can suffice for fine-tuning, we further explored whether its position within the network affects performance: the test performance remains consistent regardless of the unfrozen layer's position. Even when only the first layer (immediately after the embedding layer) is unfrozen, it can still influence the subsequent layers, enabling the model to produce informative embeddings for the regression head at the final layer.

**Transformer Parts Ablation**. To investigate which components of the Transformer architecture contribute most to performance, multiple models were trained with specific parts of the last two layers unfrozen. These include feed-forward layers, attention mechanisms, and individual components of the attention module—key, query, value, and output projection layers. Performance (cf. Figure 9) increases progressively, starting from the embedding layer, followed by key, query, value, and output projections, then the feed-forward and attention layers, and finally, the full last two layers.

This analysis suggests that ESM2 does not encode mutation-specific knowledge in individual layers, as it does for structural features such as contacts and binding sites (Vig et al., 2020). Fine-tuning performance is largely invariant to the position or number of fine-tuned layers, indicating that adaptation likely arises from task-specific tuning of the overall embeddings rather than mutation-specific mechanisms. Notably, the differences observed across Transformer components demonstrate the parameter efficiency of multi-head self-attention, which achieves competitive performance with approximately half the parameters of the feed-forward layers.

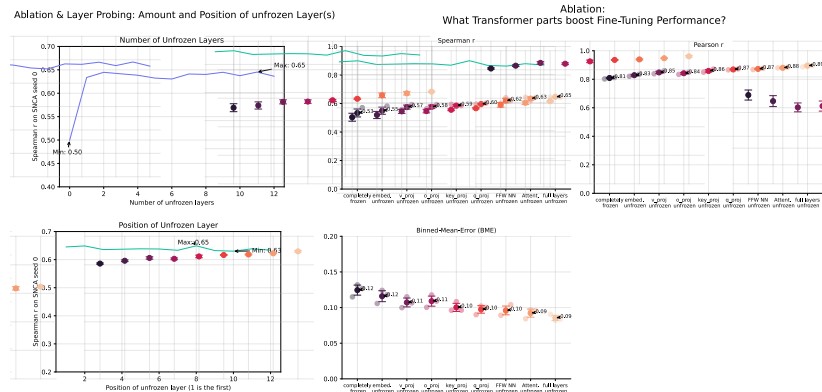

Figure 9: Ablation study of ESM-Effect: Fine-Tuning and Layer probing. Ablating Transformer parts of optimized ESM-Effect on 3 SNCA seeds.

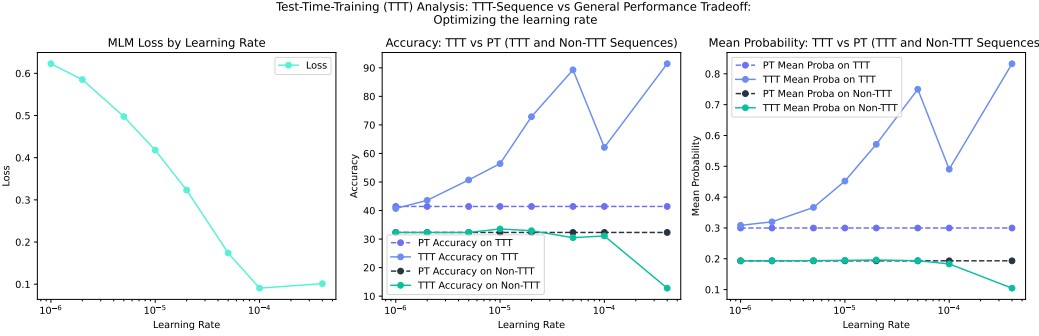

Figure 10: Customizing ESM2 backbone on SNCA sequence while maintaining general knowledge and preventing catastrophic forgetting.

## 7.3 EXPERIMENTS WITH TEST-TIME-TRAINING (TTT)

As Bushuiev et al. (2024) showed, fine-tuning a pretrained PLM backbone on a specific protein sequence that is used for a given inference task improves performance (Bushuiev et al., 2024). For instance, unsupervised mutation pathogenicity prediction from PLMs without a regression head benefited from TTT. Here, we sought to apply this technique to ESM-Effect using a similar approach for supevised functional: first we customize (i.e. fine-tune) the ESM2 backbone on the protein sequence of the DMS. Then we train the backbone with the ESM-Effect head on top on a DMS. To customise the 35M ESM2 model, we started with the hyperparamters recommended by Bushuiev et al. (2024) However, this led to rapid overfitting to the DMS sequence: for the target DMS sequence and another non-DMS related sequence, we monitored the percentage of correctly predicted tokens and their probability when predicting the each token in the sequence individually (with a mask for that token). We used this strategy to adjust the learning rate to maintain accuracy of the non-related sequence while achieving increased accuracy on the TTT/DMS sequence (cf. Figure 10). Based on the results we selected 1e-5 as optimal, customized the ESM2 backbone and trained ESM-Effect on three seeds of the SNCA DMS.

Experiments with SNCA (seeds 0–2) reveal only minor performance differences between the non-TTT and TTT models, depending on the metric used. Consequently, no significant benefit from TTT is observed in this setting.

## 7.4 GENERALIZATION TEST

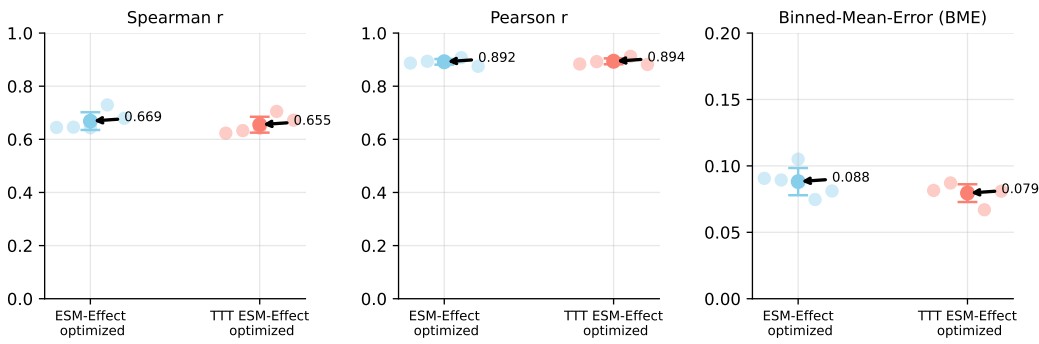

Figure 11: Customizing ESM2 backbone on SNCA sequence while maintaining general knowledge and preventing catastrophic forgetting.

To investigate to what extent ESM-Effect might learn features from one member of a protein family that may allow it to generalize to other family members we trained ESM-Effect on the Glucokinase DMS (with 20% test split) and evaluated its performance on the test split and on a second DMS from the SRC tyrosine kinase (Ahler et al., 2019).

First, we analyze the difference between the two DMS: we counted frequencies for each of the $19^{19}$ wildtype - mutant amino acid pairs to investigate distributional shift bias. The frequencies are dependent on the relative frequency of the respective wildtype amino acid in the sequence but also whether the experimental readout for the mutation succeeded. The cosine similarity of the two frequency matrices is 0.88 and Spearman rho is 0.62 suggesting that DMS-specific mutation frequencies may only have a mild impact on generalization. Second, we investigated the distribution of the catalytic activity scores (cf. Histogram Figure 3).

After min-max scaling the SRC DMS scores to the range of GCK DMS scores, we compare the two matrices with the mean catalytic activity score for each wildtype-mutant amino acid pair

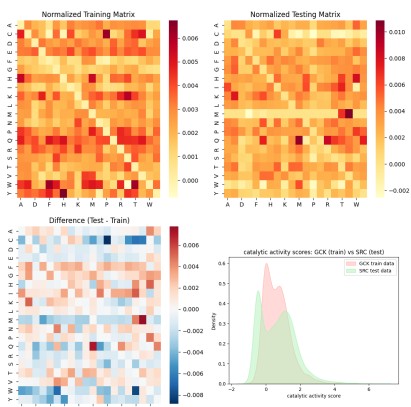

Figure 12: Matrices comparing the mean catalytic activity scores for all wildtype residue - substituting amino acid pairs between the train (GCK) and test (SRC) data. Histogram comparing the catalytic score distributions for the Glucokinase training DMS and the SRC kinase testing DMS. This shows that the I.I.D. assumption does not hold true anymore. Accordingly, ESM-Effect performs poor

finding that they are fairly distinct: although cosine similarity is still at 0.736, Spearman correlation is 0.1.

The histogram in Figure 3 underscores that the **two DMS represent two completely different distributions**, which is biologically plausible: even though both are kinases, their binding pocket and catalytic domain are fairly distinct as they process completely different substrates. Thus, we expect generalization to be poor. And indeed **generalization is very poor**: there is almost no correlation between predictions and ground truth scores (Spearman rho 0.03) despite training on a kinase DMS (Figure 4).

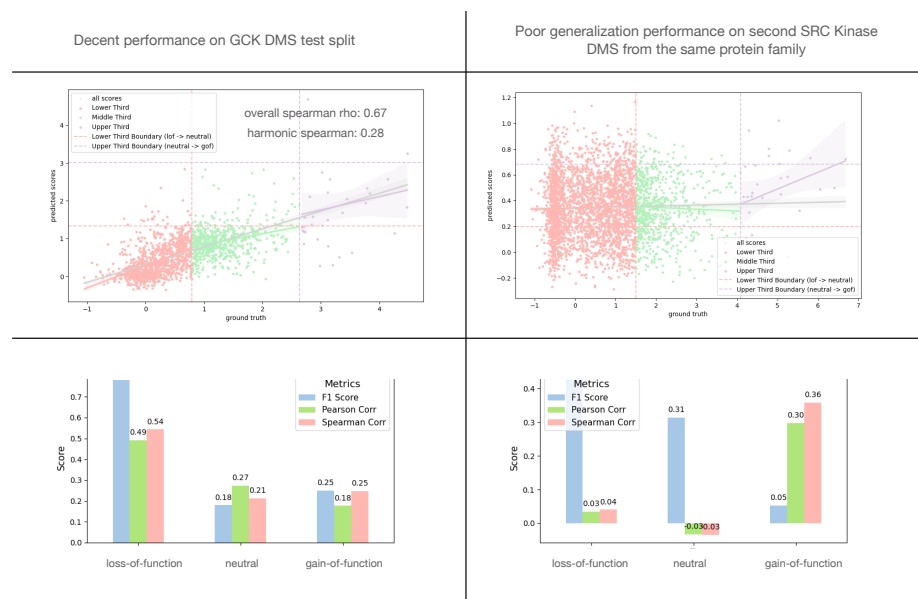

Figure 13: ESM-Effect was trained on 80 percent of mutation from the GCK DMS. Left column shows performance on 20 percent testing data versus poor performance when evaluating generalization from the Glucokinase to the SRC tyrosine kinase. The three different colors and regression lines represent the respective thirds of the score range corresponding to the three effect classes (LoF, Neutral and GoF). The overall Spearman rho for the test split is 0.67 and the Harmonic Spearman is 0.28

## 7.5 DISSECTING THE NOTION OF A WILDTYPE SEQUENCE

Over the course of ongoing evolution many different variants of sequences evolve and are selected for fitness. Thus, one fixed, unique "wild-type" sequence does not exist. Only different versions of sequences exist which have different properties. The term "mutation" and "variant" build on the arguable existence of one unique, static "wild-type" sequence in which one amino-acid is substituted forming the mutant sequence. Nonetheless, a physiological, natural sequence space exists comprising many functionally and fitness-regarding equivalent "wild-type" sequences which are curated in databases like UniProt (The UniProt Consortium et al., 2023), UniRef or SwissProt (Suzek et al., 2007; Boeckmann, 2003). These databases typically list one fixed, reference/"wild-type" sequence but also other isoforms. And different amino acid alterations in these physiological sequences may be viewed as mutations in contexts like precision medicine, where the wildtype sequence (space) for a given oncogene is established. In this light, the task of variant pathogenicity prediction equates to carving out the edges of the physiological sequence space. So the notion of one unique wild-type sequence is less applicable to variant pathogenicity prediction models, since the models learn a notion of physiological sequence spaces to which they compare a given sequence at inference. Yet they require a reference sequence (one version of the physiological wildtype) to compare the likelihood of the variant amino acid to: There is no effect without a reference to compare the effect to. The same applies to supervised, specialists models trained on DMSs. While we train models that only take the mutated sequence as input to predict the DMS score, the DMS score itself is being calculated by comparing the enrichment of the cell expressing the mutant sequence to cells expressing the reference sequence. In general, variant prediction is not possible without a reference sequence (as part of the physiological sequence space).

## 7.6 EMBEDDING ANALYSIS

Seeking to understand how fine-tuned ESM2 embeddings compare to baseline ESM2 embeddings - the reason ESM-Effect outperforms PreMode - we obtained the embeddings for 100 GCK DMS test mutations from both models and analyzed them using the UMAP dimensionality reduction technique. However, there are no clusters and coloring the data points according to their catalytic activity does not show any relationship either. This might be due to the regression head's role of extracting meaningful features from the embedding (as it is trained with an order-of-magnitude higher learning rate) or due the UMAP assumption of a uniform distribution not holding true for ESM2 embeddings as they are probably not uniformly distributed across the entire manifold but rather form clusters.

## 7.7 TRAINING MEAN EMBEDDING MODELS

Fine-tuning ESM2 models with a regression head using the mean sequence embedding presents unique challenges that do not arise when using mutation position embeddings. Notably, these issues are specific to the PTEN stability and enzyme activity DMS datasets and are not observed for SNCA or NUDT15. Training with the mean embedding often exhibits instability, characterized by spiking losses and abrupt fluctuations in performance. Additionally, convergence is slow, requiring more than 20–30 epochs, because the mean embedding condenses information from many model parameters into a lossy representation, making it harder for the model to capture fine-grained mutation effects. Furthermore, the gradients from the regression head propagate less directly through the mean operation to the ESM2 model, compared to using the mutation position embedding, where the gradients flow directly from the head to the relevant model parameters. This instability mainly applies to fine-tuning the full ESM2 model on the PTEN enzyme activity DMS compared to frozen or LoRA-based models. Therefore the PTEN enzyme activity comparison in Figure 2 is lacking given our limited compute resources in order to train enough models for enough epochs in the fully unfrozen setups.

## 7.8 METHODS

### 7.8.1 TRAINING

We don't fine-tune all parameters of the model but freeze the top 10 of 12 layers for the 35M model and split the learning rate: ESM2 parameters are updated by 1e-5 and prediction head parameters by 1e-4 times the local batch size. We use gradient accumulation for larger batches with a local batch size of 4 and 2 accumulation steps. Dropout rate was set to twenty percent and we train for 10 epochs with a one cycle learning rate scheduler. AdamW was used with $\beta_1 = 0.9$, $\beta_2 = 0.999$, $\epsilon = 1e-8$ and weight decay coefficient = 0.01. Training time for a DMS with 6k mutations for 10 epochs is roughly $1\frac{1}{2}$ up to 2 hours on a NVIDIA L4 GPU depending on evaluation and monitoring.

### 7.8.2 DATA

We used the same DMSs as in PreMode to compare performance: the exact same 20 percent test split with five different seeds was used for random splitting. Note that when using data from the PreMode repository the same csv file contains scores for all properties of the DMS if there are multiple. As the score column names are not indicative of the measurement, and the same measurement type has different score column indices for different datasets we specify them here:

We used the same amount of unfreezed ESM2 backbone weights and did not adjust the capacity of the model to the size of the dataset. To evaluate generalization from training on GCK we use a DMS of the SRC kinase from MAVEDB containing 3372 mutations (Ahler et al., 2019; Rubin et al., 2021). To adjust the scale of the score measurement from the SRC DMS to GCK DMS we use min-max scaling. Code is available in the following GitHub repository: `https://github.com/lovelacecode/ESM-Effect`.

| Protein | column name for enzyme activity |
|---------|--------------------------------|
| SNCA | score.1 |
| CYP2C9 | score.1 |
| NUDT15 | score.2 |
| CCR5 | stability: score.1 |
| | binding: Abd7: score.2 |
| | binding HIV-1: score.3 |
| ASPA | score.2 |
| GCK | score.1 |
| PTEN | score.2 |

Table 2: Mapping of proteins to column names containing enzyme activity scores.

