# OpenReview forum: "ESMGain: Effective and Efficient Prediction of Mutation’s functional Effect via ESM2 Transfer Learning and robust Benchmarks"
_ICLR.cc/2025/Conference — Submitted to ICLR 2025_

### Official Review · Reviewer_N19Z · 2024-10-16

**Soundness:** 2
**Presentation:** 1
**Contribution:** 2
**Rating:** 3
**Confidence:** 4

**Summary:**

The paper introduces a method for mutation effect prediction. The method relies on two ESM2 heads for generating protein sequence embeddings, one used with wildtype sequences and the other one for the mutated sequences. On top of the embeddings a custom regression head is trained. The technical novelty of the method is their design of the regression head and the fact that the two ESM2 models have different weights one fine-tuned for wildtype sequences and the other for the mutated sequences. Other contributions claimed by the paper are towards better bencharking (i) testing generalization of the models fine-tuned on one protein by testing them on a different protein from the same family and (ii) introduction of “Harmonic Spearman” as a new metric.

**Strengths:**

1. The generalization test is of interest. In Figure 3, authors show the different distribution of labels for two different proteins from the same family and convincingly show why generalization between proteins (even in the same family) is not easy.

**Weaknesses:**

1. Paper is poorly structured, making it very hard to read:

	- Introduction contains contents which would better fit to related work or background (“Notably, PreMode was pre-trained to predict the binary measurement of “pathogenicity” for 4.7 million mutations and uses AlphaFold2 predicted protein structure, Multiple Sequence Alignments (MSAs) and pre-trained ESM2 650M embeddings as features (John Jumper, 2021).”) And it also presents some results and their discussion (“That leads us to hypothesize that the signal provided by protein structure, MSAs, and embeddings is largely redundant for the task of effect prediction. PreMode’s ablation studies show minimal performance drop when any of these modalities is ex- cluded, suggesting that they capture overlapping information for functional effect prediction. This explains ESMGain’s superior performance in turn: its fine-tuned embeddings are task-specific and the single modality avoids the redundancy.”). I suggest honoring the usual structure of the paper and using introduction just for motivation and a very brief (not so detailed) teaser for the contributions of the paper.

    - Chapter 4, which should be describing the technical novelty and the method does not provide that many details, for example Figure 1 illustrating the method is never referenced in the text. I suggest to use a figure and equations to better describe the regression head, instead of the textual description at the end of section 4.2.

    - No table summarizing results. The reported numbers are scattered across text and some figures, making it very hard to get a glimpse of the results. I suggest a more transparent summarization of the results, such as by using a table.

2. Poor formatting of the paper.

    - Authors are not economical with the space by being sometimes too verbose, repetitive in repeating their contributions or for example by wasting the whole first page just on abstract. Being more economical would enable the authors to make bigger figures which have too small fonts and are hard to read. I suggest making figure large enough so the fonts can be legible.

    - References are poorly formated. Some references starting with “…”. AlphaFold referenced as “(John Jumper, 2021)” - note that AlphaFold was a collective effort. I suggest proper citing and formatting of references.

3. Insufficient literature survey. Authors only have 13 references. I suspect authors were trying to fit into the page limit of 10 pages including references - this is not necessary references dont count in the page limit. I suggest making proper literature survey and crediting relevant work. For example, I miss the reference to ProteinGym, arguably one of the most influential benchmarks in this area.

4. Insufficient benchmarking. Authors only focus on the comparison to PreMode (which was still not peer reviewed) and only compare on 5 proteins. I suggest to compare for example to AlphaMissense as well.

5. The key contribution of having separate ESM2 heads for wildtype and for the mutated sequence is questionable. Authors claim this to give them the key improvement by the underlying inductive bias. To me it is not clear how to decide what is wildtype and what is mutation. What if the mutation is adopted by evolution and becomes the “new wildtype” and then gets mutated again? There is no fundamental reason to distinguish between the sequences. So I believe that using the distinction between the sequences based on the dataset definition and then adapting the two heads to this definition only leads to overfitting to the dataset, potentially explaining any benefit gained from these separate heads. I dont have a concrete suggestion how to prove authors point, because I think the point is wrong. If authors stand by their point they should present convincing evidence supporting that their “inductive bias” is not just overfitting to the dataset definition of what is wildtype and what is mutation.

6. The model seems to improve over PreMod on just 2-3 out of 5 proteins (Figure 2), this does not seem very convincing. My suggestion would be to get other datasets (maybe something relevant could be found in ProteinGym) and show improvement on other dataset as well.

7. The Harmonic Spearman is just introduced at the end of the paper and not motivated well enough. Could authors explain the choice of using harmonic average? Could authors clearly compare harmonic spearman to normal spearman? How does it change the evaluation of all the benchmarked models? A table summarizing the results (as suggested in Weak point 1) would help.


I suggest to reject this paper for the following reasons. (i) The paper is not is well placed in literature, comparison to AlphaMissense is missing and the survey of the related work is not sufficient. (ii) The key contribution of using two separate ESM2 models for the wildtype and the mutated sequence is questionable and the claim of bringing a useful inductive bias is not supported by strong evidence, the improvement coming from this choice might be due to overfitting to the dataset definition of what is mutant and what is original sequence. (iii) The results dont seem as strong, only showing improvement for 2-3 out of 5 proteins. More convincing evaluation using other dataset would be necessary. (iv) The technical novelty of separate fine-tuning of two ESM models with a custom regression head is limited. (v) The writing is poor, making it hard for the reader to asses the contributions, the results of the method and its placement in the literature.

**Questions:**

Most of my questions and suggestions for the authors are listed alongside the weaknesses in the above section.

To sum up:

- I suggest to find more datasets of other proteins from ProteinGym and use them to compare ESMGain at least with PreMode and AlphaMissense.

- Present clearly the results in a table using both the Spearman correlation and the proposed Harmonic Spearman at one place.

- Authors should provide evidence for bringing a new useful inductive bias by taking the two separately fine-tuned ESM2 heads. Other evidence than the improved performance (for some proteins), which might just hint at overfitting to the dataset definition.

- The paper should be rewritten focusing on clear presentation of the method and results, clear structure of the paper and proper formatting of figures and references. The method (in particular the new regression head) is not clearly presented, the structure is chaotic with discussion of results and related work appearing already in introduction. The figures have tiny fonts making them hard to read and the references are wrongly formatted.

---

> ### Author Response · Authors · 2024-11-30
> **Addressing your concerns (part 1/2)**
>
> Thank you for reviewing our paper, we appreciate the time you have taken for thorough review! We completely rewrote the paper and conducted numerous additional analyses to address your concerns: we added a survey of existing models to clarify ESMGain's (now termed ESM-Effect) application subfield, as well as the step-by-step development of the proposed model through consecutive ablations and re-engineered the benchmarking metric.
>
> Our revisions concerning the **Weaknesses** you mentioned:
>
> - **1 & 2 paper structure and formatting:** As you suggested, we have rewritten the manuscript to address you concerns. This includes proper organization of the introduction and adding a table summarizing the results. Figure 3 depicts the model architecture (incl. the regression head) in detail. We increased the fontsize of the figures for better readability, cut redundancies and formatted the references properly.
>
> - **3 literature survey:** Thank you for pointing this out. Now we list all relevant references for pathogenicity and existing functional effect predictors as well as Benchmarks etc. Figure 1 visualizes our survey of existing approaches by comparing the two fields of pathogenicity and functional effect predictions and highlights the tradeoff between generalist models like AlphaMissense or NLR-ESM and specialist models like ESM-Effect and PreMode.
>
> - **4 Insufficient Benchmarking:** The literature survey of functional effect predictors yields two approaches based on static ESM1 embeddings and multimodal PreMode as the most advanced approach. In the newly added development process section we integrate and compare/ablate concepts of the two existing approaches and distill the most performant feature into ESM-Effect. Thus the only remaining comparison is PreMode, to which we compare in the Results section. We extended the comparison by almost doubling the number of DMS. As the patterns of outperformance (by varying margins) are consistent and our compute resources are limited, we did not compare on further DMS. A comparison to AlphaMissense is not reasonable as it provides generalist, proteome-wide pathogenicity predictions while ESM-Effect is fine-tuned on a specific DMS to produce high-accuracy functional effect predictions. To illustrate this, we compared AlphaMissense predictions to four different DMS finding that the predictions are non-informative except for one DMS without Gain-of-Function mutations (cf. Appendix Figure 1).
>
> - **5 Questionable architecture effectiveness:** Thank you for your reasonable remarks regarding the arguable existence of a (steady) wildtype sequence. Indeed, evolution adopts to fitness-enhancing mutations but this proccess happends extremely slowly and a certain physiological wildtype sequence space existis that is devoid of pathogenic sequences. Moreover, no effect is possible without a reference point to compare to: the DMS score of a sequence is calculated by comparing the enrichment of the respective sequence to the enrichment of cells with the reference sequence. Every pathogenicity predictor (incl. AlphaMissense) requires a reference amino acid to compare a given sequence (mutation) to and compare the log likelihood ratio of the mutant to a reference amino acid at a given masked position. We added a section to the Appendix relating to your concerns where we discuss the arguable existence of a wildtype sequence and its use in effect predictors. Given above points, we decided not to discard wildtype knowledge from the model, but we also investigated models with the mutant sequence as only input finding that while they perform on par for some DMS they underperform one others. Optimized ESM-Effect (only mutant embedding but knowing the mutation position) achieves comparable performance to full ESM-Effect (one model for reference, one for wildtype sequence) on all DMS. Our detailed ablations show that while mean-embedding based models can perform on par, they take much longer to train and do not match the performance of the inductive bias regression head in some cases even when the mean-embedding based model also receives the full wildtype sequence as additional input and the inductive bias head model not, proving its effectiveness.

---

> > ### Author Response · Authors · 2024-11-30
> > **Addressing your concerns (part 2/2)**
> >
> > - **6 Extending Evaluation:** As suggested, we extended the comparison to PreMode and found consistent outperformance by varying margins on 6 of 8 proteins (9th involves antibody binding, which is a distinct task with specialized methods). Besides, the (other) novelty of ESM-Effect is that it achieves performance sup-par of a multi-modal model (incl. embeddings, AF2 structure and MSAs) although it only relies on one modality by a simple yet highly effective combination of fine-tuning and an inductive bias regression head. Although the technical novelty may not stand out the most, it is its effectiveness, not its technical details which matter in practice.
> >
> >
> > In light of these responses, we hope we have addressed your concerns. If we have left any notable points of concern unaddressed, please do share and we will attend to these points. We sincerely hope that you can appreciate our efforts on responses and revisions and thus raise your score.

---

> > > ### Comment · Reviewer_N19Z · 2024-12-02
> > >
> > > Thank you for your response.
> > >
> > > I dont see response regarding my point on Harmonic Spearman. In the updated paper it seems removed, but it is still present in Figure 13. The presentation of the paper still seems rather preliminary.
> > >
> > > I stand by my point regarding the suspicion that fine-tuning the two heads for the WT/mutant sequences brings overfitting to the dataset definition of which sequence goes to which head. If the authors want to claim a general inductive bias they should prove this convincingly. This would actually be a valuable contribution if the authors could convincingly show that the fine-tuning to mutant/WT generalizes well across datasets. The authors made an effort to argue why wildtype definition is something fundamental, but this would have to be proven by an experiment. If it is that fundamental, perhaps the pLM learns it and does not need human to be told which is which.
> > >
> > > Regarding the suggestion to use ProteinGym, raised by several reviewers. Did the authors use some of it in their new evaluation? If so, how much of the available data did the authors use?
> > >
> > > The paper still does not seem fit for publication, therefore I maintain my score.

---

> > > > ### Author Response · Authors · 2024-12-02
> > > >
> > > > Thank you for your feedback. See below for our remarks addressing your concerns:
> > > >
> > > > - **Harmonic Spearman replaced by relative Bin-Mean Error:** Indeed, we have removed the Harmonic Spearman and replaced it with the relative Bin-Mean Error, which uses the Error of the prediction instead of spearman rho and uses equal-width binning instead of thirds. Thereby it weighs the model's performance on clustered, numerous and simple-to-predict mutations (often neutral) equal to the performance on rarer, non-clustered functionally active mutations which are of greater interest. And thanks for pointing out, we will remove the Harmonic Spearman from Figure 13. How can we further improve the presentation of the paper?
> > > >
> > > > - **Wildtype vs. Mutant Sequence:** In our step-by-step development, we also train models using only the (mean of the) mutant embedding (i.e. without any information of the wildtype). These perform on par with the inductive bias models (i.e. information of the position of the mutation but without wildtype residue) on the Deep Mutation Scans (DMS) from SNCA and NUDT15 but underperform on PTEN stability & enzyme activity for which the inductive bias of the mutation position holds value. This shows that the inductive bias is not mere overfitting. The optimized ESM-Effect model only contains the embedding of the mutant sequence at the mutation position (not the full wildtype sequence incl. the wildtype amino acid) and performs on par with the full ESM-Effect model (uses full wildtype and mutant sequence). How can we provide you with further evidence of the inductive bias head's value?
> > > > By Masked-Language Modelling Pretraining the pLM learns the physiological sequence space because the sequences in its trainig set (from UniRef) contain multiple isoforms per protein and evolutionarily similiar sequences (i.e. physiological mutations) but are carefully monitored to exclude malign/non-physiological sequences. Thus, ESM2 indeed knows whether a given (mutant) sequence is similar to sequences from pretrainig or not. However, one needs one of the physiological wildtype amino acids to calculate the likelihood of the variant. This is common procedure in the field and used by SOTA pathogenicity predictors such as AlphaMissense, EVE and ESM-1v.
> > > >
> > > > - **Cross-protein Generalization:** Concerning cross-protein generalization, we note in the Conclusion that current foundation models like PLMs do not learn enough about the complex biophysical reactions and interactions of proteins through masked-sequence pretraining. For instance, even between two proteins of one family the distribution of DMS scores is completely different, as is the physicochemical mechanism and substrate of the Glucokinase and Src kinase from the example in Fig.13: Glucokinase is allosterically regulated by its substrate glucose and phosphorylates it, while Src is embedded in a complex network of interactions between different signalling proteins and phosphorylates other proteins, autoinhibits and features structural domains like SH2 relevant for Gain-of-function machnisms that GCK lacks. Accordingly, robust transfer can neither be expected for ESM-Effect nor for any effect prediction model because the level of biological detail that drives the different activity of the mutations is too high for current models. Thus we moved the generalization test into the Appendix and added a intra-protein test where the model is trained and tested on mutations from different sequence intervals of the same protein which is more relevant for practical applications and within the scope of the model.
> > > >
> > > > - **Protein Gym:** The extended evaluation DMSs stem from MAVEDB while DMSs like PTEN and CCR5 are from the ProteinGym. We only used a fraction of all the available DMS in ProteinGym for evaluation, as we have limited compute resources to train 215 models with 3 seeds for PreMode and optimized ESM-Effect. Besides, our Evaluation on 9 DMS shows consistent performance of optimized ESM-Effect performing on par (2x) and above par (6x) with the only competitor for specialist DMS fine-tuned functional effect prediction, PreMode.
> > > >
> > > > In light of these responses, we hope to have addressed you concerns. We are looking forward to your reply and areas for improvement if you still identify some. Thank you for taking the time to review our submission.

---

> > > > > ### Comment · Reviewer_N19Z · 2024-12-02
> > > > >
> > > > > > In our step-by-step development, we also train models using only the (mean of the) mutant embedding (i.e. without any information of the wildtype). These perform on par with the inductive bias models (i.e. information of the position of the mutation but without wildtype residue) on the Deep Mutation Scans (DMS) from SNCA and NUDT15 but underperform on PTEN stability & enzyme activity for which the inductive bias of the mutation position holds value. This shows that the inductive bias is not mere overfitting.
> > > > >
> > > > > I am sorry I don't understand. Can you please describe this clearly as an experiment proving your inductive bias? What is the input to what model, what weights are used where, what data were used to train which head. And how does the claim that there is no overfitting follow.
> > > > >
> > > > > I apologize for not following, but your manuscript had such substantial changes that now it should be reviewed again from the beginning, it is very hard for me as a reviewer to track the changes. Regarding your question about the presentation. My tip would be to try to find a colleague who does not know the work (with fresh eyes) and ask them to kindly read your manuscript and check whether they can understand it. My eyes are not fresh for this manuscript; I am influenced by the initial submission.

---

> > > > > > ### Author Response · Authors · 2024-12-02
> > > > > >
> > > > > > ### **Question 1: Experiment/Prove for inductive bias:**
> > > > > >
> > > > > > In the step-by-step model development part (section 4.2), after ablating the ideal model size, showing that fine-tuning outperforms static embeddings, and that all fine-tuning regimen (LoRA, full FT and unfreezing layers) perform equal, we compare four different regression heads to ablate the most performant. The regression heads differ based on whether information on the mutation position is avaible or not and whether the specific wildtype amio acid (as used by the DMS) is known: We train different models with
> > > > > >
> > > > > > (1) taking only the mean embedding of the mutant sequence produced by one 10/12 layer frozen fine-tuned ESM2 model as input to a Single-Layer-Perceptron (SLP) regression head (this model only knows the mutant sequence as suggested by you),
> > > > > >
> > > > > > (2) a model with the same ESM2 backbone setup that uses the embedding of the mutant sequence at the mutation position as input into a neural network regression head (just a weight vector + SLP) (this model knows the mutation position as an inductive bias),
> > > > > >
> > > > > > (3) a model with two ESM2 backbones one producing a mutant sequence embedding, the other one a wildtype sequence embedding and which then combines the respective embeddings at the mutation position through weight vectors and a 2-LP into a single prediction. Besides, we compared a model
> > > > > >
> > > > > > (4) similar to (3) but use the mean of the entire-sequence embedding of wildtype and mutant sequence as input to the same regression head to investigate the value of using the entire embedding (pooled by the mean).
> > > > > >
> > > > > > We evaluate these models for 3 seeds on the DMS PTEN stability and enzyme activity (from ProteinGym), NUDT15 and SNCA (from MaveDB) (cf. Figure 2.4): while all models perform on par for NUDT15 and SNCA DMS, the mean based models underperform on both PTEN DMS. Even though the input of the mean-based model (4) contains mutation position and the specific wildtype-residue (i.e. the wildtype sequence) it does not reach the performance of model 2 which proves that giving only the embedding at the specific mutation position to the regression head inducts the model effectively leading to higher performance.
> > > > > >
> > > > > >
> > > > > > ### **Substantial Changes**
> > > > > >
> > > > > > Prompted by your through review, we rewrote the paper and conducted additional analyses to further prove our points. To provide you with a better overiew of the changes: The introduction is streamlined and the Related Work section and Figure 1 contain a detailed survey of existing models for pathogenicity and functional effect prediction. Analyzing characteristics from existing PLM-based effect predictors, we now conducted a step-by-step development and ablation procedure to prove the optimal performance of all model components. Then we benchmark against multimodal PreMode and show ESM-Effect's outperformance. After that, we introduce the Benchmarking Framework, analyze ESM-Effect prediction characteristics and assess its transfer capabilities from an interval within the same DMS that is distinct from the training mutation's interval with mixed performance results. Finally, the paper ends with a conclusion emphasizing current limitations as outlined in our previous response. Thank you for your advice regarding the presentation, which we will incorporate to further refine the manuscript's coherence.

---

### Official Review · Reviewer_SFRF · 2024-10-30

**Soundness:** 2
**Presentation:** 2
**Contribution:** 2
**Rating:** 3
**Confidence:** 4

**Summary:**

The paper proposes a novel model method  ESMGain for predicting the functional impact of protein mutations, expanding addressing limitations in existing binary pathogenicity predictors. By fine-tuning ESM2 embeddings with a custom regression head, ESMGain aims to accurately classify mutations as loss-of-function, neutral, or gain-of-function. Through evaluations  in catalytic activity prediction tasks, ESMGain outperforms the state-of-the-art baselines by leveraging only ESM2 embeddings. Besides, the authors propose a new benchmarking framework for functional effect prediction, emphasizing cross-protein generalization tests within the same protein family. A Harmonic Spearman metric is also introduced to balance performance evaluation across mutation effect categories.

**Strengths:**

1) The paper is well-structured and clearly written.

2) The proposed method achieves state-of-the-art performance on selected datasets in functional effect prediction.

3) By employing two independent ESM2 models to embed wildtype and mutant sequences separately, the paper addresses potential information loss in mutation representation, enhancing the model’s ability to capture subtle differences. Ablation studies demonstrate that only using ESM2 embeddings effectively captures most of the relevant information on DMS datasets,  effectively reducing the reliance on additional data modalities.

4) The paper proposes a novel benchmarking framework for functional effect prediction incorporates a cross-protein generalization test within the same protein family.

**Weaknesses:**

1) The novelty of this paper is limited. The use of dual ESM2 embeddings to separately represent wildtype and mutant sequences, along with the introduction of the Harmonic Spearman metric to address label imbalance, appears more incremental than groundbreaking.

2) It seems that the motivation of the proposed benchmarking framework is underdeveloped. While focusing on cross-protein generalization within the same family is technically interesting, it lacks a clear connection to real-world situations where this type of evaluation would be essential.

3) While ESMGain performs well on the tested DMS data, its generalization to other samples within the same protein family is weak (cross-family tests). The model may be overfitting in the specific training proteins.

**Questions:**

1) The paper includes relatively few citations and offers limited analysis of related work. Could the authors clarify if this indicates that the approach is less informed by recent research developments?

2) Additional questions are noted in the "Weaknesses" section.

---

> ### Author Response · Authors · 2024-11-30
>
> Thank you for reviewing our paper, we appreciate the time you have taken for thorough review! We completely rewrote the paper and conducted numerous additional analyses to address your concerns: we added a survey of existing models to clarify ESMGain's (now termed ESM-Effect) application subfield, as well as the step-by-step development of the proposed model through consecutive ablations.
>
> Our revisions concerning the **Weaknesses** you mentioned:
>
> - **1 limited novelty:** While many efforts have focused on pathogenicity prediction, the clinically more relevant but biologically more challenging field functional effect prediction lacks (1) a survey of existing methods, (2) comparisons to pathogenicity predictors in order to prove performance gains obtained and (3) realistic metrics to asses progress compared to previous methods and pathogenicity predictors as well as Benchmarking strategies that are more biologically realistic than random splits of the dataset. Our paper includes a comprehensive survey (cf. Figure 1) and comparisons of AlphaMissense to DMS data (cf. Appendix Figure 1) demonstrating the limited applicability except for one DMS lacking biologically significant Gain-of-Function mutations and thereby justifying the need and opening up the subfield of functional effect prediction. Functional Effect prediction is a relatively young field and few methods and no benchmarks exist. This paper contributes with a robust benchmarking metric, proposing standardized datasets and testing regimens on distinct sequence intervals. Besides, we collate features of existing approaches, perform detailed, step-by-step performance comparison and ablation developing the ESM-Effect model. We then prove its effectiveness by comparing it to competitor PreMode and show that ESM-Effect outperforms PreMode on 6 of 9 DMS even though it lacks AF2 structure and MSA input. This is an important finding which contrasts the benefit from multi-modality observed in natural language processing. The conclusion emphasizes that the field of AI x biology needs new pretraining objectives and data about the complex physicochemical reactions and interactions of proteins as evidenced by respectable yet far from ideal performance on the challenging task of functional effect prediction.
>
> - **2 relevance of cross-protein transfer:** The motivation of the cross-protein generalization test is the overall goal of the field to provide highly accurate predictions for all possible proteins and their functional properties which is still far-away. As such transfer capabilites are not realistic for current models, we revised the generalization test by testing on separate sequence intervals of the DMS protein instead of using random splits (previous strategy) or a distinct DMS (cf. Figure 7, Section 5.4). As you pointed out, this is more relevant to practical applications.
>
> - **3 transfer capability:** Indeed, the cross-transfer capability in the same sub-family (Kinases) is weak. However, this is to be expected for specialist models like ESM-Effect and PreMode which provide higher accuracy but are limited to one specific DMS. Even between two proteins of one family the distribution of DMS scores is completely different, as is the physicochemical mechanism and substrate of the Glucokinase and Src kinase: Glucokinase is allosterically regulated by its substrate glucose and phosphorylates it, while Src is embedded in a complex network of interactions between different signalling proteins and phosphorylates other proteins, autoinhibits and features structural domains like SH2 relevant for Gain-of-function machnisms that GCK lacks. Accordingly, robust transfer can neither be expected for ESM-Effect nor for any effect prediction model because the level of biological detail that drives the different activity of the mutations is too high for current models. Thus we moved the generalization test into the Appendix and added a intra-protein test where the model is trained and tested on mutations from different sequence intervals.
>
> Concerning your **question**:
>
> As we misunderstood the ten page limit to include references, we now added all relevant references to the revised form including all previous methods for functional effect prediction from whose strengths and weaknesses we develop ESM-Effect in a detailed ablation and development proccess. Additionally we added a survey of existing methods for generalist pathogenicity prediction, hybrid models also applicable to DMS (yet with low accuracy) and specialist functional effect predictors (cf. Figure 1).
>
> In light of these responses, we hope to have addressed your concerns successfully. If we have left any notable points of concern unaddressed, please do share and we will attend to these points. We sincerely hope that you can appreciate our efforts on responses and revisions and thus raise your score.

---

### Official Review · Reviewer_i8jy · 2024-11-04

**Soundness:** 2
**Presentation:** 2
**Contribution:** 2
**Rating:** 3
**Confidence:** 3

**Summary:**

This work proposed a method called ESMGain to use fine-tuning ESM2 with a custom regression head incorporating inductive biases and enable the application of learned protein semantics to functional effect prediction. This method outperforms state-of-the-art competitor PreMode on deep mutational scans from three different enzymes.

**Strengths:**

1. The proposed method performs the best for functional effect prediction in the dataset.
2. The methodology of ESMGain can predict functional effects without the limitation of feature redundancy and task specificity.

**Weaknesses:**

1. The organization needs improvement. Some terms like "PTEN" didn't have full names. The size of font in those figures is too small to read and is not consistent. The section 7 should be in the section of the experiential setup.
2. In Fig4, "LoF, Neutral and GoF" in captions should be the same as the text in x axis of figure. How about the performance in all other baselines like competitor PreMode in Fig4?
3. Have you conducted multiple train-test split seeds in ablation study of ESMGain? Why the result of the original ESMGain in the ablation study is different from the one in Fig2? Do they use different datasets or strategies to train and test?

**Questions:**

see above. The questions are about the description of the result and additional result in other baselines.

---

> ### Author Response · Authors · 2024-11-30
> **Extensive Revisions regarding identified Weaknesses**
>
> Thank you for reviewing our paper, we appreciate the time you have taken for thorough review! We completely rewrote the paper and conducted numerous additional analyses to address your concerns: we added a survey of existing models to clarify ESMGain's (now termed ESM-Effect) application subfield, as well as the step-by-step development of the proposed model through consecutive ablations.
>
> Our revisions concerning the Weaknesses you mentioned:
>
> - **1 improving organization:** we ensured that special terms and abbreviations are clearly explained and text on the revised figures is large enough and in a consistent font. We re-organized the structure of the paper for conciseness and clarity. Please point out any remaining inconsistencies in the revised manuscript if you still notice some.
>
> - **2 remark on the generalization test:** as pointed out by another reviewer and we have now explained in the introduction, the generalization to a different DMS is not the primary objective of specialist functional effect predictors like ESM-Effect and PreMode. Thus we moved Figure 4 to the Appendix and replaced it with a generalization test to distinct sequence regions within the same DMS which is more relevant to practical applications.
>
> - **3 questions on ablation study:** We have significantly expanded the ablation study, and integrated it as model development process which proves each property of ESM-Effect (i.e. model size, fine-tuning, regression head) step-by-step to be most effective through detailed comparison/ablation on 4 DMS with 3 seeds respectively. Fine-Tuning and the inductive bias regression head stand out as key improvements over existing approaches.
>
> In light of these responses, we hope we have addressed your concerns successfully. If we have left any notable points of concern unaddressed, please do share and we will attend to these points. We sincerely hope that you can appreciate our efforts on responses and revisions and thus raise your score.

---

### Official Review · Reviewer_KQf9 · 2024-11-08

**Soundness:** 2
**Presentation:** 2
**Contribution:** 2
**Rating:** 3
**Confidence:** 4

**Summary:**

This paper introduces a method for fine-tuning protein language models, specifically ESM2, using deep mutational scanning (DMS) data. The fine-tuning process involves generating both local and global representations of the reference and mutant protein sequences by utilizing separate, mostly frozen ESM models for the two sequences. These representations are combined and passed through a two-layer linear neural network to predict quantitative measurements from a DMS assay.

Further, the authors propose two modifications to the evaluation of fine-tuned models. First, they recommend fine-tuning models on one protein and testing them on a different protein within the same family, rather than using held-out positions from the original protein. Second, they suggest calculating correlation metrics separately for LoF, neutral, and GoF mutations. These separate correlation scores are then combined using a harmonic mean to produce a single protein-level metric.

**Strengths:**

1. The authors introduce important ideas for better evaluating fine-tuned models: (a) evaluating models on completely held out proteins and (b) developing a metric that prioritizes performance on LoF and GoF variants over neutral variants.
2. Their fine-tuning approach demonstrates superior performance compared to existing methods, such as PreMode and augmented versions of unsupervised models.
3. Through ablation studies, the authors establish that using larger versions of ESM2 does not significantly improve performance and that employing separate models for reference and mutant sequences provides some benefits.

**Weaknesses:**

1. Limited dataset evaluation: The authors do not evaluate their method on the large compendium of DMS datasets that are available in ProteinGym (217 datasets covering 2.5 million mutations), instead focusing on only 5 datasets (Figure 2). To convincingly prove that their fine-tuning approach outperforms existing methods, they should expand their analysis to more datasets.

2. Insufficient comparison to existing fine-tuning approaches: PreMode and augmented unsupervised models are not the only approaches that have been proposed to fine-tune protein language models on DMS datasets. See https://www.nature.com/articles/s41467-024-51844-2 and https://arxiv.org/pdf/2405.06729. These papers explore strategies such as parameter-efficient fine-tuning and fine-tuning jointly on multiple DMS assays that this paper does not consider. In particular, the approach proposed in the second paper listed above shows improved performance on entirely held out proteins, which is in stark contrast to the poor generalization to new proteins exhibited by ESMGain in Fig. 4.

3. While the idea to compute separate correlation metrics for LoF, neutral, and GoF variants is clever, the method of dividing variants into these categories by splitting the ground-truth scores into thirds is arbitrary. A more robust method, such as a Gaussian mixture model with three components, could provide a more principled assignment of variants to these classes.

**Questions:**

1. The poor generalization performance in Fig 4 to new proteins seems to indicate that ESMGain is overfitting. Can you more heavily regularize your model to avoid this?
2. Do you find that performance depends on what fraction of ESM2 is frozen? What happens if it is entirely frozen and you only train a 2-layer NN on top of the reference/mutant representations?
3. Does the harmonic Spearman correlation provide a more meaningful ranking than say AUROC at distinguishing the bottom third from the top third of variants?

---

> ### Author Response · Authors · 2024-11-30
> **Comprehensive Revision addressing your concerns (part 1/2)**
>
> Thank you for reviewing our paper, we appreciate the time you have taken for thorough review! We completely rewrote the paper and conducted numerous additional analyses to address your concerns: we added a survey of existing models to clarify ESMGain's (now termed ESM-Effect) application subfield, as well as the step-by-step development of the proposed model through consecutive ablations and re-engineered the benchmarking metric.
>
> Our revisions concerning the **Weaknesses** you mentioned:
>
> - **1 limited dataset evaluation:** we almost doubled the number of comparisons to SOTA competitor PreMode, now encompassing all relevant DMS on which PreMode was tested as well. While PreMode was evaluated on further stability DMS, we consider stability prediction to be a different field from DMS prediction, for which separate, non-DMS datasets exist with tailored stability prediction approaches. Besides, our limited compute resources did not allow to run ESM-Effect on all 217 DMS and our extended evaluation on 9 DMS demonstrated consistent evidence of outperformance (6 of 8 non-binding DMS). Additionally, the novelty is that ESM-Effect outperforms PreMode despite lacking AF2 structure and MSAs as input.
>
> - **2 Insuffcient comparison to existing approaches:** Thank you for sharing these interesting papers. The first paper shows the benefit of fine-tuning PLMs and uses DMS predictions as one example: it shows that LoRA fine-tuning of ESM models with a simple SLP(mean mutant embedding) head outperforms the (weak) baselines Reference-Free-Analysis and homology based modeling on DMSs of AAV, GFP and GB1. We now also explore LoRA and the SLP(mean mutant embedding) head in the development/ablation of ESM-Effect showing that LoRA performs on par with Full fine-tuning and unfreezing the last two layers. Additionally, our inductive bias prediction head outperforms the mean regression head.
> The ESM-NLR approach from the second paper aims to improve the application of the protein language model ESM-1v to pathogenicity (not functional effect) prediction by fine-tuning on 25 DMS. Thus, ESM-NLR is a generalist model and pathogenicity predictor which is benchmarked on ClinVar and DMS. In the added survey we underscore that the benefit of ESM-NLR is its generalist nature being applicable to all DMS yet achieving low accuracy (cf. the predition plots in the Appendix of ESM-NLR and the comparison to AlphaMissense in our Figure 1). ESM-NLR achieves comparable performance to AlphaMissense, which we show in the Appendix is not useful for functional predictions rather for general benign vs. pathogenic distinctions. ESM-Effect however is a specialist model like fine-tuned PreMode achieving high accuracy on one specific DMS. These predictions are still of value as DMS typically span only specific domains and contain missing values from failed mutagensis.
>
> - **3 Weaknesses of the harmonic spearman:** We understand that the division into thirds seems arbitrary to you. While a gaussian mixture model with three components seems helpful, we realized that the labels for GoF, Neutral and LoF have to be calculated with hard borders themselves and the continuos nature of DMS score distributions should be respected by the metric. Thus we replace the harmonic spearman by developing the relative Bine-Mean Error (rBME) which uses equal-width binning to align with the continuous nature of the distribution. Instead of correlation which is useful for pathogencitiy prediction, we now use the absolute difference (i.e. error) of the predictions compared to the DMS label scores. The equal width binning treats performance for clustered neutral mutations and for rare unclustered gain-of-function variants equally yielding a more accurate metric. We also added a figure to visualize the calculation process which underscores why the binning is so effective. Thus, we hope that this revision addresses you concerns.

---

> > ### Author Response · Authors · 2024-11-30
> > **Comprehensive Revision addressing your concerns (part 2/2)**
> >
> > Our revisions concerning your **Questions**:
> >
> >  - **1 poor generalization to distinct DMS:** As explained for weakness 2, ESM-Effect is a specialist functional effect predictor like PreMode etc. As such they are not designed for effective transfer but for high accuracy on the specific DMS. This does not show that ESM-Effect is overfitting but rather that neither existing approaches enable highly accurate predictions on any protein's DMS with any property as a limitation of today's AI modelling capabilities. Thus, we remarked the need for new pretraining objectives and data for foundation models that understand the physicochemical reactions of the enzymes and protein interactions to realise  extremely effective and accurate functional effect predictors in the future. Besides, we moved the generalization test on a separate DMS to the Appendix and introduced testing the model on distinct sequence intervals instead of a random split as used for the PreMode comparison because this is more relevant for practical applications. The distinct interval tests show strong and weak results depending on the interval.
> >
> > - **2 number of frozen layers and static embeddings:** In the added step-by-step ablation/development of ESM-Effect we scrutinise whether fine-tuning improves over static embeddings (i.e. your suggestion of entirely freezing the ESM2 backbone) and find fine-tuning to significantly outperform static embeddings which are a key limitation of previous methods for functional effect prediction (Derbel et al. Marquet et al and PreMode by Zhong et al.). Regarding the number of layers frozen, we added an analysis to the Appendix showing that the numer of trainable layers has no considerable influence on performance (as long as at least one layer is trainable). We also found that different positions of one single unfrozen layer in the ESM2 backbone yield equal performance.
> >
> > - **3 Harmonic Spearman vs AUCROC for distinguishing mutations** Could you please elucidate what you mean by your question? How should the metric help to distinguish testing instances? As we have replaced the Harmonic spearman with the rBME, we hope that rBME addresses you concern also regarding the arbitrariness of splitting the scores into thirds. Please let us know, if you identify further aspects for improvement of the metric and its capabilities.
> >
> > In light of these responses, we hope we have addressed your concerns. If we have left any notable points of concern unaddressed, please do share and we will attend to these points. We sincerely hope that you can appreciate our efforts on responses and revisions and thus raise your score.

---

### Meta-Review · Area_Chair_YvPy · 2024-12-20

**Metareview:**

The paper considers the problem of functional effect prediction and develops an ESM2-based framework that is tailored to this task through ablation studies.

The paper provides interesting insights such as the conclusions drawn from the ablation studies and the resulting framework outperforms comparison approaches. However, all reviewers agree that the approach is of limited novelty. It is therefore critical to provide a high-quality evaluation, where none of the results can be questioned and where all conclusions are clearly presented.

The reviewers made excellent suggestions, e.g. concerning potential bias resulting from the use of separate ESM2 heads, regarding generalization test, regarding scores, etc. Towards this goal, the authors have made substantial revisions to the manuscript, which require a new cycle of reviews.

**Additional Comments On Reviewer Discussion:**

The key points raised by the reviewers were lack of novelty, which cannot be changed at this point; organization and clearer presentation of the experimental results, potential inductive bias of the approach, and the need to improve benchmarking (e.g. w.r.t. ablations, which scores are being reported etc. The authors have made substantial changed to their manuscript to address these points, but the changes are so substantial that the paper needs to go through a new submission cycle to be assessed fairly in its present form.

---

### Decision · Program_Chairs · 2025-01-22

Reject